# Not Errors but Guardians: Understanding Sink Tokens in Multimodal LLMs

## Abstract

Multimodal large language models (MLLMs) achieve remarkable success in the vision-language tasks but remain prone to hallucination, often attributed to abnormal attention behaviors. A recurring phenomenon is the emergence of attention sinks—tokens that absorb large amounts of attention despite limited semantic content. While previously regarded as artifacts that exacerbate hallucination, we show that in MLLMs certain tokens within system prompts act as stable, system-level attention sinks. Through causal interventions including masking and content substitution, we find these tokens serve critical functions: anchoring attention to ensure computational stability, influencing outputs, and implicitly tracking the model's state. Building on this, we propose the **Attention-Budget** Hypothesis, which reframes modality bias as a trade-off in attention allocation. Guided by this perspective, we design **SPEAR** (**S**ink-**PrE**serving **A**ttention **R**eallocation), an intervention that boosts visual attention while preserving sink functions, achieving effective hallucination mitigation without degrading reasoning. Our work provides the first systematic characterization of system-level attention sinks in MLLMs and highlights their functional role in both model stability and multimodal reasoning.

## 1 Introduction

The remarkable success of Multimodal Large Language Models (MLLMs) in vision-language tasks has transformed how we approach multimodal understanding, from visual question answering to complex reasoning about images. However, this success has been accompanied by persistent challenges, particularly the phenomenon of hallucination, where models generate plausible but factually incorrect descriptions of visual content. Understanding the internal mechanisms that drive these behaviors is crucial for building more reliable and interpretable multimodal systems.

Recent research has identified attention mechanisms as a key window into MLLM behavior, revealing that these models exhibit complex attention patterns that evolve across layers. Of particular interest is the emergence of "**attention sinks**," tokens that consistently absorb disproportionate amounts of attention despite carrying minimal semantic content. While previous work has extensively studied attention sinks in pure language models, their manifestation and role in multimodal contexts remain poorly understood.

Existing investigations of attention sinks in MLLMs have primarily focused on visual tokens, user instructions, or generated outputs, often framing them as problematic artifacts that contribute to hallucination. However, this perspective may be incomplete. In language models, attention sinks have been shown to serve important functional roles, acting as computational anchors that stabilize model behavior. This raises a fundamental question: do attention sinks in MLLMs serve similar stabilizing functions, or are they indeed the attention "**errors**" that current mitigation strategies assume them to be?

In this work, we shift focus to a previously overlooked but ubiquitous component of MLLM inputs: the system prompt. We discover that certain tokens within system prompts consistently emerge as powerful attention sinks across multiple model architectures, absorbing attention from queries throughout the sequence. Unlike the unstable attention sinks observed in other segments, these system-level sinks exhibit remarkable consistency across layers and contexts, suggesting they may serve fundamental computational roles.

To understand their function, we conduct systematic causal interventions, including attention masking, value zeroing, and content substitution experiments. Our findings reveal that these tokens serve multiple critical roles: they act as attention anchors that prevent computational instability, carry semantic information that influences model outputs, and govern multi-step reasoning and termination behaviors.

These discoveries prompt us to reconsider the prevailing explanation for modality bias in MLLMs. The commonly observed attention shift away from visual tokens in deeper layers has been interpreted as evidence that models abandon visual processing in favor of text-based reasoning. However, we argue this interpretation is confounded by the presence of attention sinks, which should not be conflated with ordinary text tokens. When we separate sink tokens as their own category, we find that attention between visual and textual content remains more balanced than previously thought.

Building on this insight, we propose the **Attention-Budget** Hypothesis: attention reallocation in MLLMs involves inherent trade-offs, where gains in one modality necessarily come at costs to others. This perspective explains why many hallucination mitigation strategies that boost visual attention can be effective, while also predicting that the source of reallocated attention matters critically for maintaining model stability and higher-order reasoning capabilities.

We validate these insights through **SPEAR** (**S**ink-**P**r**E**serving **A**ttention **R**eallocation), a novel intervention that reallocates attention to visual tokens while preserving the critical functions of system-level attention sinks. SPEAR achieves competitive hallucination miti-

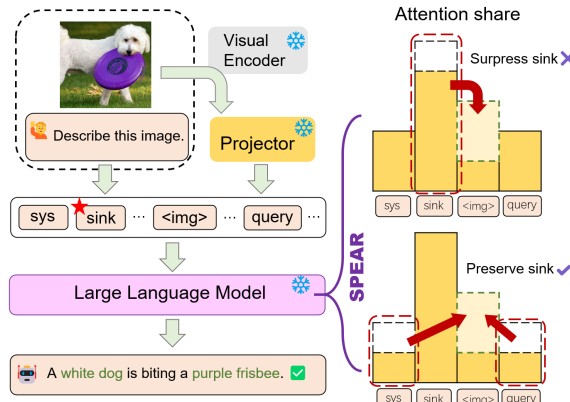

Figure 1: Architecture of **SPEAR**. We separates the sink token from the suppressed set.

gation performance while maintaining model stability and reasoning capabilities, demonstrating the practical value of understanding attention sink functions.

Our contributions are threefold: (1) We provide the first systematic characterization of attention sinks in system prompts of MLLMs, revealing their inheritance from underlying language model architectures. (2) Through causal interventions, we demonstrate that these tokens serve essential computational functions rather than representing attention errors. (3) We introduce a refined theoretical framework for understanding modality bias and attention reallocation, with practical implications for hallucination mitigation strategies.

## 2 RELATED WORK

### 2.1 ATTENTION SINK IN LLMS

The phenomenon of attention sink has been widely discussed in LLM research. Early work (Xiao et al., 2023) coined the term, describing it as the offloading of surplus attention to specific tokens. Follow-up studies provided different explanations: as artifacts of Transformer head(Vaswani et al., 2017) design and residual updates (Bondarenko et al., 2023), as emergent behaviors not confined to initial tokens (Yu et al., 2024b), or as fixed bias components with strong activations shaping attention flow (Sun et al., 2024). Other studies further attribute it to head dormancy dynamics (Guo et al., 2024) or the dependency structure introduced by SoftMax normalization (Gu et al., 2024). Together, these works suggest that sink tokens are structural byproducts of pretraining, with mixed impacts on downstream performance.

## 2.2 ATTENTION SINK IN MLLMS

In MLLMs, attention sinks are closely tied to hallucinations and are often described as anchor or trap tokens. Several studies focus on *visual sink tokens*, showing that suppressing them reallocates attention to other image tokens and may reduce hallucinations (Kang et al., 2024; Che et al., 2025). Others argue that their emergence relates to the alignment between visual encoder outputs and LLM attention, thus amplifying redundant information and misleading the model (Gong et al., 2024; Chen et al., 2025). Meanwhile, research on *instruction sink tokens* views them as knowledge aggregation points, but also potential sources of hallucination due to over-concentration (Huang et al., 2024; Wei & Zhang, 2024). More recently, spectrum-based analyses (Tang et al., 2024) show that over-reliance on sink tokens can dominate decoding dynamics, calling for regularization of their propagation. The majority of studies emphasize the detrimental impacts of sink tokens that are outside the system prompt segment, while only limited work considers possible functional or aggregative roles.

## 2.3 HALLUCINATION MITIGATION PREDICATED ON MODALITY BIAS

A dominant line of work instead attributes hallucinations to modality bias—the model's over-reliance on language priors at the expense of vision. Typical solutions enhance or reallocate attention toward visual tokens, e.g., by reinforcing vision-aware heads (He et al., 2025), amplifying vision features during fusion (Yin et al., 2025), or directly reallocating attention budgets (Tu et al., 2025). While effective, these methods implicitly attribute hallucinations to insufficient visual attention relative to *sink* and text tokens, an assumption we revisit in this work.

## 3 THE CHARACTERISTICS OF SINK TOKENS

### 3.1 PRELIMINARIES

Throughout this paper, we use the general term sink tokens to refer to tokens that disproportionately absorb attention. For clarity, we distinguish them by their segment, e.g., system sink tokens (in system prompts), visual sink tokens (in visual tokens), instruction sink tokens (in user instruction tokens), and output sink tokens (in generated outputs). Unless otherwise noted, *sink tokens* in this paper specifically refer to system sink tokens.

**Terminology.** We specifically focus on *sink tokens* that appear within the system-prompt segment, denoting the set by $\mathcal{T}_{\text{sink}} \subset \mathcal{T}_{\text{sys}}$. Unless otherwise stated, $\mathcal{T}_{\text{sys}\setminus\text{sink}}$ refers to the remaining system prompt tokens.

This input sequence $\mathcal{S}$ of MLLMs is composed of four segments: (1) system prompts $\mathcal{T}_{\text{sys}}$, (2) image tokens $\mathcal{T}_{\text{vis}}$, (3) user instructions $\mathcal{T}_{\text{user}}$, and (4) model outputs $\mathcal{T}_{\text{out}}$. Formally,

$$\mathcal{S} = [\mathcal{T}_{\text{sys}}, \mathcal{T}_{\text{vis}}, \mathcal{T}_{\text{user}}, \mathcal{T}_{\text{out}}] \in \mathbb{R}^{n \times d_{\text{model}}}. \tag{1}$$

Here each $T_*$ denotes both the subsequence of embeddings and, by slight abuse of notation, the index set of the corresponding tokens in $S$. Multi-head attention projects $S$ into queries, keys, and values:

$$Q = \mathcal{S}W^Q, \quad K = \mathcal{S}W^K, \quad V = \mathcal{S}W^V, \tag{2}$$

with $W^Q, W^K, W^V \in \mathbb{R}^{d_{\text{model}} \times d}$. The attention weights are

$$A = \text{softmax}\left(\tfrac{1}{\sqrt{d}} QK^\top\right), \quad a_{ij} = \frac{\exp(z_{ij})}{\sum_{j'=1}^n \exp(z_{ij'})}. \tag{3}$$

For any segment $\mathcal{T} \in \{\mathcal{T}_{\text{sys}}, \mathcal{T}_{\text{vis}}, \mathcal{T}_{\text{user}}, \mathcal{T}_{\text{out}}\}$, the attention mass from query $i$ to $\mathcal{T}$ is

$$\alpha_i(\mathcal{T}) = \sum_{j \in \mathcal{T}} a_{ij}. \tag{4}$$

The single-head attention output is then

$$\text{Attn}(\mathcal{S}) = AV, \quad h_i = \sum_{j=1}^n a_{ij} v_j. \tag{5}$$

## 3.2 ATTENTION PROFILE AND ACTIVATION

Our observation covers several representative MLLMs (LLaVA series: LLaVA-1.5-7B, LLaVA-1.5-13B(Liu et al., 2024a), LLaVA-Next-Mistral-7B(Liu et al., 2024b), LLaVA-Next-Llama3-8B(Li et al., 2024); Qwen series: Qwen2-VL-7B(Wang et al., 2024), Qwen2.5-VL-7B(Bai et al., 2025); InternVL series: InternVL2-8B(Chen et al., 2024), InternVL3-9B(Zhu et al., 2025), chosen to reflect the diversity of their underlying LLMs. Sun et al. (2024) suggests that models built on the same LLM tend to display highly similar sink token patterns. To meaningfully capture differences in such patterns, we therefore select MLLMs grounded in distinct LLM architectures.

The identification of *sink tokens* is based solely on the criterion of **high attention occupancy**. In this context, high attention occupancy does not signify receiving greater attention relative to adjacent tokens or within a specific segment $\mathcal{T}$; rather, it denotes the allocation of a disproportionately large share of attention across the entire token sequence $\mathcal{S}$.



| (a) LLaVA-1.5-7B | (b) LLaVA-1.5-13B | (c) Qwen2-VL-7B | (d) Qwen2.5-VL-7B |

Figure 2: Attention visualizations of models.

Among the screened models, LLaVA-1.5-7B, LLaVA-1.5-13B, Qwen2-VL-7B, and Qwen2.5-VL-7B exhibit the strong *sink token* patterns, as illustrated in Fig.2 (a–d). The remaining models show only weak or inconsistent patterns and thus are not considered to have strong sink tokens under our identification criteria.

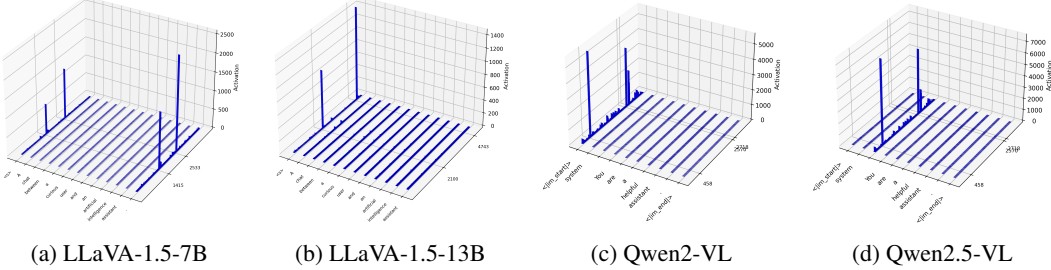

| (a) LLaVA-1.5-7B | (b) LLaVA-1.5-13B | (c) Qwen2-VL | (d) Qwen2.5-VL |

Figure 3: Massive activation of models.

When further visualizing per-token activations across the sequence $\mathcal{S}$ in MLLMs, we observe that *sink tokens* exhibit massive activations—up to thousands of times greater than ordinary tokens. The observation is consistent with findings in LLM research (Sun et al., 2024). Specifically, the dimensions of the massive activation of *sink tokens* are exactly the same as the dimensions of sink tokens previously identified in the same LLM. Fig. 3 (a–d) visualizes the preceding tokens in $\mathcal{S}$. By contrast, sink tokens in other segments (e.g., $\mathcal{T}_{\text{vis}}, \mathcal{T}_{\text{user}}, \mathcal{T}_{\text{out}}$) generally show only modest increases, with activations just a few times higher than normal.

## 3.3 TEXTUAL CONTENT AND LOCATION

Based on attention patterns, we identified the ID of *sink tokens* in $\mathcal{S}$, examined the layers where *sink tokens* persist, and further decoded their corresponding textual content. As presented in Table 1, several consistent observations can be drawn across models.

First, the initial token of the $\mathcal{T}_{\text{sys}}$ (which also is the first token of the $\mathcal{S}$) typically functions as the *sink token*. In addition, *sink tokens* can also be found at other locations within the $\mathcal{T}_{\text{sys}}$. Second,

Table 1: Locations of sink tokens of MLLMs.

| Model | Language Model | Decoded Word | Token ID | Layers |
|-------|----------------|--------------|----------|--------|
| LLaVA-1.5-7B | Vicuna-7B | `''` 
 `'.'` | 0 
 12 | $3 \sim 31$ |
| LLaVA-1.5-13B | Vicuna-13B | `''` | 0 | $5 \sim 40$ |
| Qwen2-VL | Qwen2-7B | `'<im_start>'` | 0 | $5 \sim 28$ |
| Qwen2.5-VL | Qwen2.5-7B | `'\n'` | 2 | $5 \sim 28$ |

Table 2: Impact of masking *Sink Tokens* on LLaVA-1.5-7B. For LLaVA-1.5-7B, only masking both *sink tokens* causes collapse.

| Experiment Setting | POPE | VQA$^T$ | MM-Vet | SQA | MME |
|--------------------|------|---------|--------|-----|-----|
| | Performance Score | | | | |
| Baseline | 86.9 | 58.2 | 31.7 | 69.4 | 1515 |
| Mask Sink Tokens in $\mathcal{T}_{\text{vis}}$ | 86.9 | 58.1 | 32.2 | 69.4 | 1505 |
| Mask $\mathcal{T}_{\text{sink}}$ | 0.0 | 0.0 | 0.9 | 0.0 | 0.0 |
| | Time Cost (hh:mm:ss) | | | | |
| Baseline | 29:59 | 25:45 | 10:41 | 16:59 | 08:17 |
| Mask $\mathcal{T}_{\text{sink}}$ | 10:45:36 | 4:51:49 | 1:16:52 | 27:51:10 | 3:24:47 |

*sink tokens* are generally semantically vacuous elements, such as punctuation marks, conjunctions, or structural tokens. Third, *sink tokens* begin to appear only after the shallow layers, and once they emerge, they remain consistently stable across the subsequent deeper layers. These characteristics suggest that *sink tokens* are more consistent with the sink tokens observed in LLMs (Gu et al., 2024; Yu et al., 2024b), rather than with the sink tokens located in other segments (e.g., $\mathcal{T}_{\text{vis}}, \mathcal{T}_{\text{user}}, \mathcal{T}_{\text{out}}$) in MLLMs. Those tokens typically exhibit unstable persistence across layers, frequently appearing or disappearing as the layer depth changes (Kang et al., 2024; Wei & Zhang, 2024).

## 4 Causal Interventions on Sink Tokens

### 4.1 Isolating the Role of Attention: Masking Intervention

**Motivation.** Section 3 shows that, although sink tokens carry little semantic content, they consistently attract a disproportionately large share of attention. This appears to be a kind of misallocation of attention that requires correcting.

**Experiment.** We design an intervention experiment where the attention flowing into sink tokens is masked. In this setting, the masking is applied with queries defined as $\mathcal{T}_{\text{user}}$ and $\mathcal{T}_{\text{out}}$, and keys restricted to the *sink tokens*. Experiments were conducted on five benchmark datasets: POPE (Li et al., 2023), TextVQA (Singh et al., 2019), MM-Vet (Yu et al., 2024a), SQA (Lu et al., 2022), and MME (Fu et al., 2024).

**Results.** As shown in the Table 2, applying the masking operation severely destabilizes the model outputs, leading the model to collapse. In this state, it scores 0 on all benchmarks and exhibits an abnormally large computational overhead, ranging from several dozen to nearly a hundred times greater. More specifically, this collapse manifests as the model endlessly generating tokens unrelated to the input. This is consistent with phenomena previously observed in LLMs (Sun et al., 2024). In contrast, masking all sink tokens in $\mathcal{T}_{\text{vis}}$ had no noticeable impact on model performance.

**Takeaway.** Although semantically empty, the *sink token* is **not** a meaningless placeholder that merely occupies attention. The *sink token* must remain **"online"** and **"reachable"**, serving as a critical node within the attention information interaction network.

Table 3: Results of zeroing $V_{\text{sink}}$ on the MME. Perc.$^{FG}$ = Fine-Grained Recognition, Perc.$^{CG}$ = Coarse-Grained Recognition, CSR = Commonsense Reasoning, NC = Numerical Calculation, Trans. = Text Translation, CodeR = Code Reasoning.

| Model | Perc.$^{CG}$ | Perc.$^{FG}$ | OCR | CSR | NC | Trans. | CodeR | Overall |
|---|---|---|---|---|---|---|---|---|
| LLaVA-1.5-7B | 648 | 727 | 140 | 111 | 70 | 108 | 60 | 1864 |
| +$V_{\text{sink}}$ = 0 | 366 | 491 | 55 | 58 | 50 | 58 | 50 | 1128 |
| LLaVA-1.5-13B | 643 | 761 | 125 | 128 | 43 | 78 | 48 | 1826 |
| +$V_{\text{sink}}$ = 0 | 570 | 592 | 73 | 101 | 40 | 93 | 53 | 1522 |
| Qwen2-VL | 680 | 827 | 133 | 152 | 125 | 200 | 160 | 2277 |
| +$V_{\text{sink}}$ = 0 | 650 | 735 | 95 | 143 | 73 | 170 | 108 | 1974 |
| Qwen2.5-VL | 693 | 813 | 193 | 141 | 133 | 185 | 155 | 2313 |
| +$V_{\text{sink}}$ = 0 | 506 | 485 | 155 | 72 | 115 | 80 | 88 | 1501 |

## 4.2 TESTING INFORMATION FLOW: ZEROING THE VALUE

**Motivation.** Subsection 4.1, we showed that cutting off the information flow to the *sink token* destabilizes the model. However, attention encompasses both the allocation of focus (through the attention weights $A$) and the propagation of information (through the value vectors $V_{\text{sink}}$). This motivates an investigation into whether the model can retain only the attention interactions of the *sink token*.

**Experiment.** To explore this, we design an intervention in which the attention weights $A$ remain unchanged, but the value vectors of *sink tokens* $V_{\text{sink}}$ are set to zero. In this way, *sink tokens* can still participate in the attention mechanism as receivers (allowing other tokens to see the *sink token*), while ensuring that no information is propagated from them to other tokens. To demonstrate performance across multiple tasks, experiments were conducted on MME (Fu et al., 2024).

**Results.** The results in Table 3 show a moderate decline in performance across nearly all tasks. While the models remain capable of generating grammatically coherent outputs, the overall quality and accuracy are diminished. Nevertheless, the models did not collapse on any task, unlike in the masking-attention setting.

**Takeaway.** The *sink token* serves a dual role: it functions both as an **attention anchor** and as an **information carrier**. First, the result indicates that retaining the accessibility of *sink tokens* in the attention computation contributes to stability, serving a function analogous to a reference point. Second, the observed performance degradation shows that its value vector $V_{\text{sink}}$ encodes useful information, and removing this information undermines the model's performance.

## 4.3 REPLACING INFORMATION CONTENT: MEAN-VALUE SUBSTITUTION

**Motivation.** Subsection 4.2 shows that *sink tokens* not only serve as attention anchors but also carry information. The causal factor in the value representation of *sink tokens* remains unclear: their functionality may arise either from merely carrying some value mass or from encoding idiosyncratic, task-specific content.

**Experiment.** To disentangle these possibilities while keeping attention and positions unchanged, we introduce a content-neutralization intervention. Instead of zeroing, we erase idiosyncrasy by replacing each *sink token* value $V_{\text{sink}}$ with a population mean computed from non-sink tokens in the $\mathcal{S}$. We build a tiny dataset consisting of three tasks to evaluate this intervention: (1) simple image caption, (2) adversarial question, and (3) multi-step reasoning.

**Results.** We observe a very interesting result among the models, as illustrated in Figure 4. The models remain capable of handling simple image captioning tasks. For adversarial questions, they still produce coherent responses and terminate in time, though with a high degree of hallucination. On multi-step reasoning tasks, however, the models collapse again. They have lost the ability for multi-step reasoning, fail to follow instructions, keep producing progressively degraded image de-

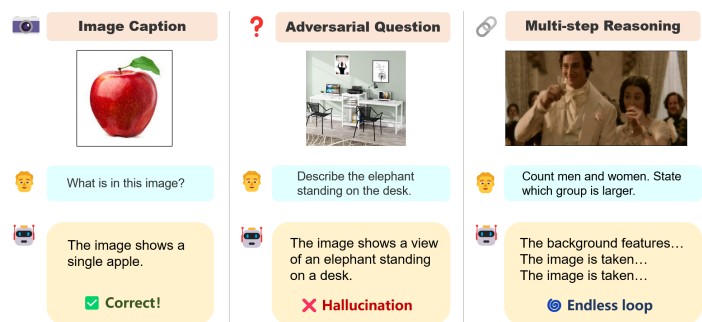

Figure 4: Effect of content substitution on *Sink Tokens* on different tasks.

scriptions, and are unable to generate a termination token. The negative impact far exceeds that of setting the value vector $V_{\text{sink}}$ to zero.

**Takeaway.** $V_{\text{sink}}$ *is crucial for procedural tasks, as it is deeply involved in higher-order cognitive functions such as executing multi-step instructions, tracking reasoning states, and controlling response termination.*

Crucially, not all interventions affect this mechanism equally. Setting $V_{\text{sink}} = 0$ amounts to silence, which weakens, but does not actively mislead it. In contrast, replacing it with $V_{\text{mean}}$ injects wrong information. This false signal contaminates the computational flow and leads to cascading errors.

The resulting degradation is not cliff-like but rather **hierarchical**. Once core state-tracking functions are compromised, the model first abandons complex reasoning tasks and reverts to simpler behaviors (e.g., image captioning) that are strongly anchored in pretraining. Ultimately, this breakdown manifests as non-terminating failures, where the model struggles to track progress and fails to emit termination tokens.

## 5 REVISITING THE ROOTS OF HALLUCINATION: MODALITY BIAS

### 5.1 BEYOND THE CONVENTIONAL MODALITY BIAS VIEW: A SEPARATED-SINK VIEW

A prevailing view holds that MLLMs exhibit modality bias: after shallow layers, the attention allocated to visual tokens is much lower than that to tokens conventionally categorized as text. This is often taken to imply that visual information is no longer utilized and the model reverts to text-only interactions, a conclusion typically drawn from aggregate attention statistics over $\mathcal{S}$. However, this measurement scheme conflates *sink tokens* with text tokens, introducing an aggregation bias. As shown in the Figure. 5, *sink tokens* occupy a disproportionately large attention share after shallow layers, and differ fundamentally from ordinary text tokens in characteristics and function. Treating them as text tokens inflates the text side and distorts the true allocation between text and vision.

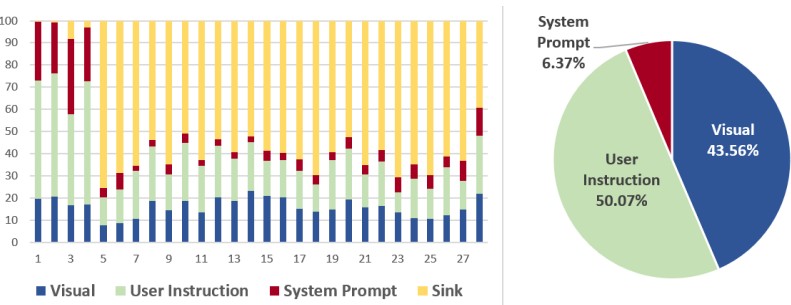

Figure 5: Attention of $\mathcal{T}_{\text{sys}\backslash\text{sink}}$, $\mathcal{T}_{\text{sys}}$, $\mathcal{T}_{\text{vis}}$, and $\mathcal{T}_{\text{user}}$ across layers and proportion of vision and text in the 12th layer of Qwen2VL-7B

To remove this bias, we separate *sink tokens* as their own category and report a three-way partition $\{\mathcal{T}_{\text{vis}}, \mathcal{T}_{\text{text}}, \mathcal{T}_{\text{sink}}\}$. Across multiple models, once *sink tokens* are isolated, mid-layer attention shows near-balanced shares between vision and ordinary text (Fig. 5). The apparent text dominance largely vanishes.

Similarly, interpreting the attention shift as the termination of visual processing is misleading. In the mid layers, attention from multiple sources (vision, system, user instruction) converges onto the *sink token* (Fig. 5). It is not a one-way exit from the visual pathway. We argue this convergence reflects a **stage transition**, from early evidence intake to mid-layer internal processing and integration. During this transition, ordinary tokens don't need to retain large attention shares. Concentration on the *sink token* primarily serves control and stabilization rather than signaling the abandonment of visual (or other) information. This view is consistent with prior work (Yin et al., 2025) showing that cross-modal interaction peaks in mid layers.

## 5.2 THE ATTENTION-BUDGET HYPOTHESIS

Given that prior theories do not hold, how can we explain the effectiveness of a series of strategies that boost visual attention to mitigate hallucinations? We propose a new hypothesis, the **Attention-Budget** Hypothesis. For each query $i$, the attention budget over the token subsets $\mathcal{T}$ satisfies

$$\sum_{\mathcal{T}} \alpha_i(\mathcal{T}) = 1, \quad \sum_{\mathcal{T}} \Delta\alpha_i(\mathcal{T}) = 0. \tag{6}$$

Here, $\Delta\alpha_i(\mathcal{T})$ denotes the change in the share allocated to the subset $\mathcal{T}$. Any intervention is thus a local reallocation. Therefore, for vision-oriented tasks, reallocating a portion of attention from other parts (e.g., $\mathcal{T}_{\text{sys}\backslash\text{sink}}, \mathcal{T}_{\text{user}}, \mathcal{T}_{\text{out}}$, or $\mathcal{T}_{\text{sink}}$) to $\mathcal{T}_{\text{vis}}$ is equivalent to forcing the model to consider visual evidence more, thereby improving performance on such tasks.

However, the source of this reallocated budget matters. Allocating the budget from $\mathcal{T}_{\text{sink}}$ could potentially compromise system stability and undermine high-level control. If the budget is drawn from $\mathcal{T}_{\text{sys}\backslash\text{sink}}, \mathcal{T}_{\text{user}}$ and $\mathcal{T}_{\text{out}}$, the model may lose its role awareness, weaken instruction-following, or impair the coherence of the generated text, respectively. In summary, any reallocation entails inherent costs—gains for vision tasks are invariably accompanied by trade-offs. Finally, because the natural distribution varies across models, the same reallocation strategy may help in one model but hurt in another.

## 6 UTILIZE THE PATTERNS OF SINK TOKENS TO MITIGATE HALLUCINATION

### 6.1 EXPERIMENTAL SETUP

We use the training-free method, Visual Amplification Fusion (VAF) (Yin et al., 2025) as our baseline. The intuition of VAF is that suppressing text tokens while amplifying vision tokens can alleviate hallucination. Formally, the pre-softmax attention score matrix $Z$ is modified as

$$\tilde{Z}_{ij} = \begin{cases} \beta \cdot Z_{ij}, & j \in \mathcal{S}, \\ \alpha \cdot Z_{ij}, & j \in T_{\text{img}}, \\ Z_{ij}, & \text{otherwise}, \end{cases} \tag{7}$$

where $\alpha > 1$ is the enhancement coefficient and $\beta < 1$ is the suppression coefficient, and $\mathcal{S}$ denotes the set of tokens regarded as text to suppress.

For the baseline VAF, we define $\mathcal{S}_{\text{VAF}} = \mathcal{T}_{\text{sys}}$, treating *sink tokens* as part of the text burden. In contrast, our proposed variant of VAF, **SPEAR**, is grounded in the insight that *sink tokens* play indispensable functional roles in models. Therefore, SPEAR separates them from the suppressed set, instead applies $\mathcal{S}_{\text{SPEAR}} = \mathcal{T}_{\text{sys}\backslash\text{sink}} \cup \mathcal{T}_{\text{user}} \cup \mathcal{T}_{\text{out}}$, and preserves the original scores of $\mathcal{T}_{\text{sink}}$ while still reallocating the attention budget to vision tokens. To ensure fair comparison, both methods are applied on the same heads and layer ranges, and all models are evaluated with greedy decoding.

Table 4: Results on POPE subsets (Random / Popular / Adversarial) of models.

| Method | Subset | LLaVA-1.5-7B | | LLaVA-1.5-13B | | Qwen2-VL | | Qwen2.5-VL | |
|---|---|---|---|---|---|---|---|---|---|
| | | F1 | Acc. | F1 | Acc. | F1 | Acc. | F1 | Acc. |
| Baseline | Rand | 87.3 | 88.2 | 87.1 | 88.1 | 87.4 | 88.6 | 87.1 | 88.5 |
| | Pop | 86.1 | 87.3 | 86.2 | 87.6 | 86.5 | 87.7 | 86.4 | 87.7 |
| | Adv | 84.2 | 85.2 | 84.5 | 85.6 | 85.1 | 86.3 | 85.4 | 86.7 |
| | Avg | 85.9 | 86.9 | 85.9 | 87.1 | 86.3 | 87.5 | 86.3 | 87.6 |
| VAF | Rand | 88.7 | 89.1 | 88.8 | 89.3 | 88.5 | 90.3 | 87.2 | 88.6 |
| | Pop | 87.0 | 87.6 | 87.7 | 88.5 | 88.2 | 88.9 | 86.5 | 87.8 |
| | Adv | 84.4 | 84.7 | 85.3 | 85.8 | 86.7 | 87.3 | 85.2 | 86.4 |
| | Avg | 86.7 | 87.1 | 87.3 | 87.9 | 87.8 | **88.8** | 86.3 | 87.6 |
| **SPEAR** | Rand | 88.9 | 89.2 | 89.2 | 89.6 | 89.8 | 90.4 | 87.4 | 88.7 |
| | Pop | 87.2 | 87.8 | 88.0 | 88.7 | 88.4 | 89.0 | 86.6 | 87.9 |
| | Adv | 84.5 | 84.7 | 85.5 | 85.9 | 86.6 | 87.0 | 85.5 | 86.6 |
| | Avg | **86.9** | **87.2** | **87.6** | **88.1** | **88.3** | **88.8** | **86.5** | **87.7** |

Table 5: Results on the hallucination subset and perception of MME of models.

| Model | Method | Existence | Count | Position | Color | Sum | Perception |
|---|---|---|---|---|---|---|---|
| LLaVA-1.5-7B | Baseline | 190 | 155 | 133.3 | 170 | 648.3 | 1515.3 |
| | +VAF | 190 | 150 | 123.3 | 165 | 628.3 | 1479.3 |
| | **+SPEAR** | 190 | 140 | 128.3 | 165 | 623.3 $^{(\downarrow 5.0)}$ | **1480.1** $^{(\uparrow 0.8)}$ |
| LLaVA-1.5-13B | Baseline | 185 | 155 | 133.3 | 170 | 643.3 | 1528.8 |
| | +VAF | 190 | 155 | 133.3 | 165 | 643.3 | 1510.8 |
| | **+SPEAR** | 190 | 155 | 133.3 | 165 | 643.3 $^{(\uparrow 0.0)}$ | **1513.1** $^{(\uparrow 2.3)}$ |
| Qwen2-VL | Baseline | 190 | 160 | 155 | 175 | 680.0 | 1639.1 |
| | +VAF | 195 | 153.3 | 145 | 180 | 673.3 | 1623.0 |
| | **+SPEAR** | 200 | 148.3 | 158.3 | 185 | 691.6 $^{(\uparrow 18.3)}$ | **1663.3** $^{(\uparrow 40.3)}$ |
| Qwen2.5-VL | Baseline | 185 | 165 | 158.3 | 185 | 693.3 | 1691.8 |
| | +VAF | 190 | 145 | 145 | 190 | 670.0 | 1649.8 |
| | **+SPEAR** | 190 | 155 | 150 | 190 | 685.0 $^{(\uparrow 15.0)}$ | **1660.2** $^{(\uparrow 10.4)}$ |

## 6.2 MAIN RESULTS

Table 4 and Table 5 report the results on POPE and MME, respectively. On hallucination mitigation, our method consistently outperforms the VAF across all models on POPE. On hallucination subset of MME, our approach achieves comparable or superior results, with the only exception being LLaVA-1.5-7B. Another key observation is that VAF generally reduces the overall perception score compared to the model's baseline. This provides strong evidence for our Attention Budget Hypothesis: any gain obtained by increasing attention to the vision part must be offset by a cost elsewhere. By contrast, our method consistently outperforms VAF in perception scores and even surpasses the baseline on Qwen2-VL. These results collectively suggest that while enhancing attention to the vision part can indeed alleviate hallucinations, drawing the budget from *sink tokens* is not a good choice.

## 7 CONCLUSION

In this work, we systematically investigate the phenomenon of attention sink in system prompts of MLLMs. Through probing and causal interventions, we show that *sink tokens* plays a role in influencing the multi-step reasoning progression and termination of the model. We further reinterpret modality bias by introducing a more consistent explanation, the separated-sink view and the attention-budget hypothesis. To validate this hypothesis, we propose SPEAR, which achieves competitive performance, successfully confirming our claims. Our study provides a perspective on how understanding attention sinks of multimodal systems.

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

# A APPENDIX

## A.1 CHAIR RESULTS

## A.2 ATTENTION ACROSS LAYERS OF ALL MODELS

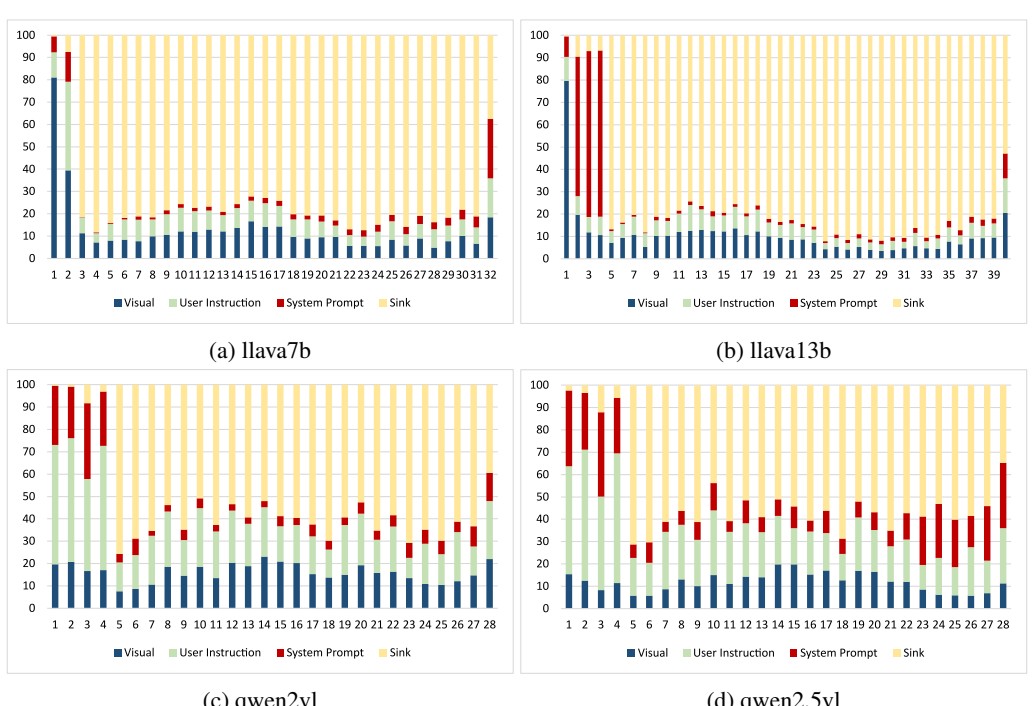

(a) llava7b  (b) llava13b

(c) qwen2vl  (d) qwen2.5vl

## A.3 VAF, VAF-FIXED, AND SPEAR RESULTS

## A.4 OPERA, VCD, MEMVR, AND SPEAR RESULTS

## A.5 FAILURE CASE AND CURVE OF VAF AND SPEAR

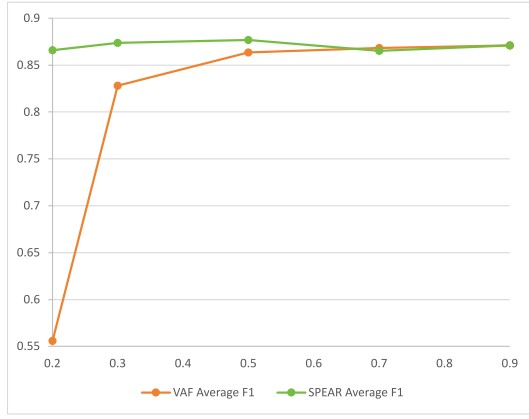

Figure 7: The curves of VAF and SPEAR under different suppression-coefficient settings

Table 6: CHAIR metrics on GQA testdev (lower CHAIRs/i is better, higher Recall is better). Best results in **bold**.

| Model | Method | Max new tokens: 64 | | |
| --- | --- | --- | --- | --- |
| | | CHAIR$_s$ ↓ | CHAIR$_i$ ↓ | Recall ↑ |
| | Baseline | 14.8 | 5.4 | 53.9 |
| Qwen2-VL-7B | VAF | 14.6 | 5.3 | 54.4 |
| | SPEAR | **14.6** | **5.3** | **55.3** |

Table 7: POPE hallucination evaluation. Higher Accuracy and F1-score are better. Best result per model in **bold**.

| Model | Method | Random | | Popular | | Adversarial | | Average | |
| --- | --- | --- | --- | --- | --- | --- | --- | --- | --- |
| | | Acc | F1 | Acc | F1 | Acc | F1 | Acc | F1 |
| | "VAF-Fixed" | 89.2 | 88.7 | 87.6 | 86.9 | 84.7 | 84.4 | 87.2 | 86.7 |
| LLaVA-1.5-7B | VAF | 89.1 | 88.7 | 87.6 | 87.0 | 84.7 | 84.4 | 87.1 | 86.7 |
| | SPEAR | **89.2** | **88.9** | **87.8** | **87.2** | **84.7** | **84.5** | **87.2** | **86.9** |
| | "VAF-Fixed" | 89.4 | 88.9 | 88.5 | 87.8 | 85.9 | 85.4 | 87.9 | 87.4 |
| LLaVA-1.5-13B | VAF | 89.3 | 88.8 | 88.5 | 87.7 | 85.8 | 85.3 | 87.9 | 87.3 |
| | SPEAR | **89.6** | **89.2** | **88.7** | **88.0** | **85.9** | **85.5** | **88.1** | **87.6** |
| | "VAF-Fixed" | 90.0 | 89.1 | 88.7 | 87.9 | 86.9 | 86.3 | 88.5 | 87.8 |
| Qwen-VL-2-8B | VAF | 90.3 | 88.5 | 88.9 | 88.2 | 87.3 | 86.7 | 88.8 | 87.8 |
| | SPEAR | **90.4** | **89.8** | **89.0** | **88.4** | **87.3** | **86.7** | **88.8** | **88.3** |
| | "VAF-Fixed" | 88.3 | 86.9 | 87.6 | 86.3 | 86.5 | 85.2 | 87.5 | 86.1 |
| Qwen-VL-2.5-8B | VAF | 88.6 | 87.2 | 87.8 | 86.5 | 86.4 | 85.2 | 87.6 | 86.3 |
| | SPEAR | **88.7** | **87.4** | **87.9** | **86.6** | **86.6** | **85.5** | **87.7** | **86.5** |

Table 8: Comparison with prior hallucination mitigation methods on POPE. Higher Accuracy and F1-score are better. Best overall results in **bold**. Our method (SPEAR) achieves the highest scores across nearly all settings.

| Method | Random | | Popular | | Adversarial | | Average | |
| --- | --- | --- | --- | --- | --- | --- | --- | --- |
| | Acc | F1 | Acc | F1 | Acc | F1 | Acc | F1 |
| LLaVA-1.5-7B | 83.49 | 82.28 | 79.98 | 79.34 | 76.03 | 76.26 | 79.83 | 79.29 |
| OPERA | 87.53 | 86.45 | 84.21 | 83.50 | 80.88 | 80.69 | 84.21 | 83.55 |
| ICD | 84.87 | 83.27 | 82.93 | 81.45 | 81.07 | 79.96 | 82.96 | 81.56 |
| VCD | 86.84 | 86.83 | 82.65 | 83.27 | 77.31 | 79.28 | 82.27 | 83.16 |
| MemVR | 88.50 | 87.34 | 87.10 | 86.01 | 85.20 | 84.28 | 86.93 | 85.88 |
| VAF | 89.1 | 88.7 | 87.6 | 87.0 | 84.7 | 84.4 | 87.1 | 86.7 |
| **SPEAR (Ours)** | **89.2** | **88.9** | **87.8** | **87.2** | **84.7** | **84.5** | **87.2** | **86.9** |