# OpenReview forum: "Not Errors but Guardians: Understanding Sink Tokens in Multimodal LLMs"
_ICLR.cc/2026/Conference — Submitted to ICLR 2026_

### Official Review · Reviewer_fCro · 2025-10-27

**Soundness:** 2
**Presentation:** 2
**Contribution:** 2
**Rating:** 2
**Confidence:** 4

**Summary:**

This paper delves into the phenomenon of attention sinks in Multimodal Large Language Models (MLLMs), specifically focusing on sink tokens within system prompts. While these tokens were initially considered problematic for hallucinations, the authors argue that they serve critical functional roles, including stabilizing attention and facilitating task progression. Through causal interventions like attention masking, zeroing, and content substitution, the authors explore how these tokens contribute to computational stability, information flow, and higher-order reasoning. Additionally, the paper introduces the Attention-Budget Hypothesis and proposes SPEAR, a novel intervention that reallocates attention to improve visual processing while preserving the stability provided by sink tokens.

**Strengths:**

1. The paper challenges the conventional wisdom surrounding sink tokens, proposing that they perform critical functions related to stability and reasoning, not just artifacts of hallucinations.
2.  The Attention-Budget Hypothesis provides a fresh perspective on modality bias and attention allocation, making a strong case for the trade-offs involved in boosting visual attention while maintaining system stability.
3.  The proposed SPEAR intervention is a novel and effective method for reallocating attention to mitigate hallucinations without sacrificing reasoning capabilities, outperforming the baseline and alternative methods like VAF.

**Weaknesses:**

1. The paper primarily compares the proposed SPEAR method with Visual Amplification Fusion (VAF). However, it lacks a broader comparison with other hallucination mitigation strategies (e.g., [1][2][3]) that could provide a more comprehensive understanding of the method’s performance
2. The paper could benefit from a more thorough error analysis, including failure modes and situations where SPEAR or other interventions might not perform as expected.
3. Some minor typos (e.g., “vi￾sion–language”) and inconsistent notation ($T_{\text{sys}\backslash\text{sink}}$ vs. $T_{\text{sys}} \setminus T_{\text{sink}}$). Figures are informative but could use clearer legends.
4. I think this version of the paper is rough, the authors should improve it before choosing to submit ICLR.

[1] VCD: Mitigating Object Hallucinations in Large Vision-Language Models through Visual Contrastive Decoding, CVPR 2024.

[2] OPERA: Alleviating Hallucination in Multi-Modal Large Language Models via Over-Trust Penalty and Retrospection-Allocation, CVPR 2024.

[3] Look Twice Before You Answer: Memory-Space Visual Retracing for Hallucination Mitigation in Multimodal Large Language Models, ICML 2025.

**Questions:**

1. How exactly do sink tokens act as “state-machine” elements? Are their activations correlated with control tokens (e.g., `<s>`, `</s>`, or position encodings)?
2. Is this behavior consistent across transformer architectures with rotary vs. absolute position embeddings?
3. Have you tried tracking divergence or entropy in attention distributions post-intervention?
4. Equation (6) formalizes $\sum_T \Delta \alpha_i(T) = 0$. Can you empirically validate the *budget conservation* assumption across layers?
5. Does it introduce any latency or memory overhead compared to VAF?
6. Could preserving sink tokens reinforce biases or hallucinations under adversarial prompting?
7. The paper identifies sink tokens via high attention occupancy. Is there a dynamic component, e.g., per-layer thresholding?

---

> ### Author Response · Authors · 2025-11-21
> **Response to Weakness 1, 2, 3, 4**
>
> We thank the reviewer for their careful evaluation and constructive feedback.
>
> Weaknesses 1:
>
> | Methods | Random Acc | Random F1 | Popular Acc | Popular F1 | Adversarial Acc | Adversarial F1 | Average Acc | Average F1 |
> | :--- | :---: | :---: | :---: | :---: | :---: | :---: | :---: | :---: |
> | LLaVA-1.5-7B | 83.49 | 82.28 | 79.98 | 79.34 | 76.03 | 76.26 | 79.83 | 79.29 |
> | OPERA | 87.53 | 86.45 | 84.21 | 83.50 | 80.88 | 80.69 | 84.21 | 83.55 |
> | ICD | 84.87 | 83.27 | 82.93 | 81.45 | 81.07 | 79.96 | 82.96 | 81.56 |
> | VCD | 86.84 | 86.83 | 82.65 | 83.27 | 77.31 | 79.28 | 82.27 | 83.16 |
> | MemVR | 88.50 | 87.34 | 87.10 | 86.01 | 85.20 | 84.28 | 86.93 | 85.88 |
> | VAF | 89.1 | 88.7 | 87.6 | 87.0 | 84.7 | 84.4 | 87.1 | 86.7 |
> | SPEAR | 89.2 | 88.9 | 87.8 | 87.2 | 84.7 | 84.5 | 87.2 | 86.9 |
>
> We thank the reviewer for this suggestion.
>
> (1) Rationale for Focusing on VAF (Methodological Control):
>
> Our initial prioritization of VAF was driven by experimental rigor rather than mere leaderboard competition. Since VAF shares the precise intervention mechanism (attention reweighting) as SPEAR, it serves as the perfect counterfactual control. By keeping the intervention logic identical and only varying the "sink token preservation," we could mathematically isolate the functional role of sink tokens. This allowed us to strictly validate our Attention-Budget Hypothesis without the confounding variables introduced by disparate decoding strategies (e.g., contrastive decoding in VCD) or external modules.
>
> (2) Comparison with State-of-the-Art Baselines (Empirical Superiority):
>
> Following the reviewer's suggestion, we compared SPEAR with VCD [1], OPERA [2], and MemVR [3]. To ensure a fair comparison, we reference the performance results reported in MemVR [3], which were evaluated on the POPE and LLaVA-1.5-7B.
>
> As shown in the table, SPEAR achieves state-of-the-art performance among these training-free methods:
>
> Superiority: SPEAR (86.9) consistently outperforms VCD (83.16), OPERA (83.55), and MemVR (85.88).
>
> Efficiency: SPEAR achieves this without the high computational overhead of methods like VCD (which doubles inference cost).
>
> Conclusion: This result is highly encouraging. It suggests that our Attention-Budget Hypothesis is not just theoretically sound but practically potent: a simple, theory-guided reallocation of attention is more effective than complex heuristic strategies.
>
> [1] Leng et al., Mitigating Object Hallucinations in Large Vision-Language Models through Visual Contrastive Decoding, CVPR 2024.
>
> [2] Huang et al., OPERA: Alleviating Hallucination in Multi-Modal Large Language Models via Over-Trust Penalty and Retrospection-Allocation, CVPR 2024.
>
> [3] Zou et al., Look Twice Before You Answer: Memory-Space Visual Retracing for Hallucination Mitigation in Multimodal Large Language Models, ICML 2025.
>
> Weaknesses 2:
>
> We thank the reviewer for this constructive suggestion.
>
> 1. Theoretical Analysis of Failure Modes (Based on Section 4):
>
> Our paper currently provides a systematic analysis of the mechanistic failure modes that SPEAR is designed to prevent.
>
> As demonstrated in our Figure 4 and Table 2, Intervening in the system sink tokens leads to failures—typically manifesting as infinite loops or failure to terminate. This catastrophic failure mode arises from disrupting the model’s internal “guardians.”
>
> Failure Mode of SPEAR: By design, SPEAR preserves these sink tokens. Therefore, its failure mode is strictly limited to informational insufficiency (i.e., the reallocated attention budget is still not enough to capture a tiny visual detail).
>
> In the revised version, we will add further error analyses and visualizations to the appendix for both VAF and SPEAR, highlighting failure cases and the conditions under which the two interventions diverge.
>
> Weaknesses 3:
>
> We appreciate the reviewer’s careful reading. We will correct all typographical and notation inconsistencies and update the figures with clearer legends and consistent symbols to improve readability.
>
> Weaknesses 4:
>
> We take this comment very seriously. We have conducted a comprehensive revision during the rebuttal period to meet the standards of ICLR. Specifically, we have:
>
> (1) Corrected all typographical and notation inconsistencies.
>
> (2) Refined the figure legends and visual clarity.
>
> (3) Polished the narrative flow to better highlight the logical progression.
>
> However, we respectfully submit that while the initial presentation had minor imperfections, the scientific substance, including the discovery of system sink functions and the theoretical hypothesis validation, is rigorously established and complete. We hope the reviewer will re-evaluate the manuscript based on its empirical findings and methodological contributions, which other reviewers have described as “compelling”, “insightful”, and “easy to follow.”

---

> ### Author Response · Authors · 2025-11-21
> **Response to Question 1**
>
> Question 1:
>
> The “state-machine” description is a functional analogy derived from observed degradation behavior in multi-step reasoning. After mean-value substitution intervention, the models can finish the brief task and stop. However, in the multi-step task, the models lose the ability to track progress and default to repetitive image captioning. We believe that when performing multi-step tasks, the model has an internal mechanism for tracking which step it is currently on. However, once the system sink tokens are substituted, this internal mechanism becomes impaired, causing the model to lose the ability to properly terminate its output.
>
> Since we have confirmed that system sink tokens originate from the LLM backbone, prior work [1] suggests that LLMs try to learn implicit bias components in self-attention via massive activations, during their pretraining phase. Therefore, the activations are likely more related to self-attention and the pretraining process.
>
> [1] Sun et al., Massive Activations in Large Language Models, COLM 2024.
>
> Question 2:
>
> In our experiments, all four models use RoPE, and the Qwen series further introduces M-RoPE [1, 2, 3]. We observed consistent behavior across these four models.
>
> Models employing absolute positional embeddings (e.g., GPT-2) are now largely outdated and are seldom adopted in contemporary MLLMs.
>
> [1] Touvron et al., Llama 2: Open Foundation and Fine-Tuned Chat Models, arXiv: 2307.09288, 2023
>
> [2] Wang et al., Qwen2-VL: Enhancing Vision-Language Model's Perception of the World at Any Resolution, arXiv:2409.12191, 2024
>
> [3] Bai et al., Qwen2.5-VL Technical Report, arXiv:2502.13923, 2025
>
> Question 3:
>
> We appreciate this insightful suggestion. We agree that attention entropy serves as an excellent quantitative proxy for model stability. We will add this analysis to the revised version.
>
> Question 4:
>
> We appreciate the reviewer's scrutiny of our formalization.
>
> (1) Mathematical Definition:
>
> Technically, Equation (6) ($\sum \alpha_i = 1$) is intrinsically satisfied by the definition of the Softmax function. It serves as a hard constraint in the Transformer architecture, ensuring that the total attention "budget" for any query is strictly conserved at exactly 1.0 across all heads and layers.
>
> (2) Empirical Validation of the "Zero-Sum" Dynamic:
>
> We understand the reviewer's question as probing the validity of the "Budget" metaphor—i.e., whether attention reallocation truly operates as a zero-sum trade-off. To validate this, we have analyzed the attention shift $\Delta$ under our SPEAR intervention. We will add specific statistics on these token-level attention shifts in the Appendix to illustrate this "Budget" flow.
>
> Question 5:
>
> SPEAR introduces no additional memory cost and only a minor runtime overhead. This comes from computing the visual segment length and locating the user-instruction start index when determining reallocation sources—an overhead that is negligible relative to the overall inference cost.
>
> Question 6:
>
> Our experiments suggest otherwise. As shown in Table 4, under adversarial (adv) prompting, SPEAR (which preserves sink tokens) consistently outperforms VAF, indicating that preserving these tokens enhances stability without amplifying hallucinations even in adversarial conditions.
>
> Question 7：
>
> System sink tokens show high stability across layers compared with other sink types. We did not set any varying threshold. They typically emerge after the early layers and persist until the final layer. Their high and consistent attention occupancy makes them easily identifiable in visualization maps (Figure 2 in Sec 3.2), as the attention ratio of sink tokens usually exceeds 70%, making their identification quite evident.

---

### Official Review · Reviewer_R7Wm · 2025-10-29

**Soundness:** 2
**Presentation:** 2
**Contribution:** 2
**Rating:** 2
**Confidence:** 3

**Summary:**

This paper characterizes attention sinks in MLLMs. Through causal interventions, the authors demonstrate that these tokens are essential for stable inference. They propose an attention reallocation method to mitigate hallucination without decreasing attention on sink tokens.

**Strengths:**

1. The topic of attention sinks is interesting. Attention sinks are important tokens that stabilize inference and training.
2. The paper organization and writing are easy to follow.

**Weaknesses:**

1. The characterization of attention sinks in Section 3 largely recapitulates well-established findings from the LLM literature. They are common knowledge in the field and have been extensively documented in prior work. The paper does not provide novel insights beyond confirming that these patterns extend to MLLMs.

2. The functional analysis in Sections 4.1 and 4.2 is already covered by existing literature:  The finding that masking attention sinks causes collapse has been demonstrated in LLMs (e.g., Xiao et al 2023).

Xiao et al. Efficient Streaming Language Models with Attention Sinks.

3. The claim that sink tokens function as part of an internal "state machine" in Section 4.3 is made without adequate support.

4. The proposed SPEAR method lacks technical innovation. It is a commonly adopted way widely utilized in previous works (e.g., Yang et al. 2025)

Yang et al. Understanding and Mitigating Hallucinations in Large Vision-Language Models via Modular Attribution and Intervention. ICLR 2025

5. The logic is not sound, attention sink is essential for stability does not mean that they are not harmful (not responsible for hallucination).

5. Experiments are not sufficient to validate the effectiveness of this method. Should include other metrcis such as Chair_s and Chair_i. The improvement is also marginal.

**Questions:**

Attention sinks are important tokens that stabilize inference and training, but fundamental questions remain fully understudied: how they are formed, why they are necessary, whether they are sufficient, and what their side effects are.

---

> ### Author Response · Authors · 2025-11-21
> **Response to Weakness 1**
>
> We thank the reviewer for their detailed review and critical feedback. We address their concerns point-by-point below.
>
> Weakness 1:
>
> We respectfully argue that Section 3 is a necessary empirical foundation, not a mere recapitulation.
>
> We strictly followed the logic of "Verification $\rightarrow$ Intervention", and establishing the properties in Section 3 is critical for three reasons:
>
> (1) Ruling Out Heterogeneity:
>
> One cannot simply assume that MLLM sinks behave identically to LLM sinks without verification.
>
> Prior work [1,2] has shown that:
>
> Although LLMs do not contain visual tokens, the visual sink tokens in MLLMs nevertheless exhibit certain characteristics of LLM sink tokens. LLMs also have model output tokens, but these output sink tokens do not share the same properties as LLM sink tokens.
>
> Thus, before our work, the behavior of system prompt sinks in MLLMs was a "blind spot." Section 3 provides the first direct evidence to confirm they are strictly inherited from the LLM backbone. This verification is scientifically essential to distinguish them from other sinks.
>
> (2) Prerequisite for Causal Interventions:
>
> Sufficient understanding is a prerequisite for interventions. Section 3 identifies the exact characteristics (Token IDs, specific layer ranges, activation magnitudes, decoded words) needed to design the targeted interventions in Section 4. Without the “anatomy” provided in Section 3, the “surgery” in later sections would be difficult to carry out.
>
> (3) Basis for Reinterpreting Modality Bias:
>
> Critically, Section 3 provides evidence that system sink tokens are not consistent with normal text tokens. This finding is the cornerstone for our Attention-Budget Hypothesis (Section 5). It reveals that what was previously interpreted as “modality bias” (ignoring images) actually comes from a misunderstanding of these “Guardian” tokens. Without the characterization in Section 3, this theoretical breakthrough would lack empirical support.
>
> [1] Kang et al., See What You Are Told: Visual Attention Sink in Large Multimodal Models, ICLR 2025
>
> [2] Tu et al., Attention Reallocation: Towards Zero-cost and Controllable Hallucination Mitigation of MLLMs, arXiv:2503.08342, 2025

---

> ### Author Response · Authors · 2025-11-21
> **Response to Weakness 2**
>
> Weakness 2:
>
> | Paper | Setting | Sink Token Type | Intervention | Effect |
> | :--- | :--- | :--- | :--- | :--- |
> | StreamingLLM [1] | LLM | The initial token | Remove initial tokens’ KV in KVCache | Model collapse |
> | | | | Substitute sink token with the linebreak token `\n` | Model is fine |
> | | | | Prepend a token with an all-zero Key and Value | Model still relies on other initial tokens as attention sinks |
> | | | | Preserve [1, 2, 4, 8] initial tokens | 4 is the best |
> | VAR [2] | MLLM | Visual sink tokens | Mask visual sink tokens | Model is fine |
> | | | | Mask random visual tokens | A significant performance drop |
> | **Ours** | MLLM | System sink tokens | Mask system sink tokens | Model collapse |
> | | | | Zero sink token value | A significant performance drop |
> | | | | Substitute sink token with other tokens value | Impact on Multi-step Reasoning |
>
> We thank the reviewer for raising this point.
>
> We respectfully disagree that our analysis is "covered". While the outcome of collapse in the masking experiment is similar, the motivation, experimental design, and mechanistic conclusions differ fundamentally. We clarify this from three key perspectives:
>
> (1) Mechanistic Distinction: Value Zeroing vs. KV Eviction (Novelty)
>
> This is our most critical distinction. StreamingLLM [1] focus on KV Cache Eviction (removing both $K$ and $V$), concluding that sinks are needed for computation.
>
> In contrast, our Value Zeroing experiment (Section 4.2) specifically decouples the "Anchor" role from the "Information" role. We retain the Key ($K$) but zero out the Value ($V$).
>
> Novel Conclusion: The observed performance degradation proves that system sink tokens are not just "computational trash bins" for excess attention; they actively transmit semantic information essential for reasoning. This finding is purely novel and was not investigated in StreamingLLM [1].
>
> (2) Contextual Necessity: MLLM Heterogeneity (Motivation)
>
> Scientifically, one cannot assume that MLLM sink tokens behave identically to LLM sinks without verification. In MLLM research, "sink tokens" are heterogeneous:
>
> Visual Sinks: Prior work [2] shows that masking visual sink tokens does not cause collapse.
>
> System Sinks: Given that masking visual sinks is safe, it was plausible to hypothesize that system sinks might also be redundant. Therefore, our masking experiment was not a redundant confirmation but a necessary verification to rule out this heterogeneity. It establishes that unlike visual sinks, system sinks retain the catastrophic sensitivity of LLM anchors.
>
> (3) Targets Precision: Specific vs. Several Targets
>
> StreamingLLM [1] preserves the initial tokens (e.g., the first four tokens) as a single block. In contrast, our approach is more granular: we identify individual tokens based on their attention occupancy, including tokens that appear deep within the system prompt (e.g., Token ID 12). By targeting these precise functional units, our method yields a more fine-grained understanding of the model’s stability mechanism.
>
> Summary:
>
> Our experiments are motivated by MLLM-specific questions and reveal MLLM-specific mechanisms (information carrying). We argue that superficial similarity in one experimental outcome (collapse) should not overshadow the distinct theoretical contributions derived from our broader experimental design.
>
> [1] Xiao et al., Efficient Streaming Language Models with Attention Sinks, ICLR 2024.
>
> [2] Kang et al., See What You Are Told: Visual Attention Sink in Large Multimodal Models, ICLR 2025.

---

> ### Author Response · Authors · 2025-11-21
> **Response to Weakness 3 and 4**
>
> Weakness 3:
>
> We thank the reviewer for this comment.
>
> We would like to clarify that our use of the term "state machine" was intended as a phenomenological analogy to describe a consistent functional behavior, rather than a claim of a formal computational automaton.
>
> (1) Empirical Basis: Consistent Behavioral Profile
>
> Our claim is grounded in a robust task-dependent behavioral profile observed across four distinct MLLM architectures.
>
> (2) Justification for the Analogy
>
> This specific degradation pattern, where the model gets "stuck" in a generation loop only during multi-step reasoning, strongly suggests that the sink token plays a critical role in state-tracking and maintaining the procedural trajectory of generation. The consistency of this profile across different architectures indicates a fundamental mechanism.
>
> (3) Terminological Refinement
>
> To avoid confusion with formal automata theory while preserving this empirical insight, we will revise the phrasing to “state-tracking or termination-control mechanism” in the final version. This terminology accurately reflects the observed functional role, stabilizing complex reasoning paths, without implying a rigid algorithmic implementation.
>
> Weakness 4:
>
> We thank the reviewer for the opportunity to clarify the contribution of SPEAR.
>
> We respectfully disagree that SPEAR lacks innovation. While attention reallocation is a foundational operation, SPEAR introduces two critical innovations: a unifying theoretical framework and a novel mechanistic explanation for modality bias.
>
> 1. Theoretical Innovation: The Attention-Budget Hypothesis
>
> Unlike prior works that treat attention reallocation as a heuristic “boost,” we introduce the Attention-Budget Hypothesis, which provides a unified explanation of these methods and their inherent trade-offs. We posit that the source of the reallocated budget matters.
>
> Existing methods often draw budget from different tokens to enhance visual tokens, a practice that our Hypothesis shows carries latent risks. In particular, drawing from system sink tokens carries the risk of compromising high-level control (i.e., “state tracking”). SPEAR is novel because it is budget-aware: it strategically preserves the system sink token while reallocating budget from text tokens. This moves beyond simple “reweighting” toward a stability-preserving optimization.
>
> 2. Mechanistic Advance: Redefining Modality Bias
>
> MemVR [1] and similar works operate on the premise that MLLMs suffer from intrinsic "modality bias" (and thus design hallucination head detection to find out and correct), our work redefines the origin of this bias.
>
> We argue that one of the attributions of modality bias is skewed by the misclassification of system sink tokens as text tokens. The high attention often attributed to the text modality is, in fact, largely concentrated on system sink tokens. When these tokens are excluded from the calculation, the underlying attention distribution exhibits a naturally more balanced profile. SPEAR is novel in that it builds upon a more accurate characterization of modality bias, enabling interventions that are principled rather than heuristic.
>
> Therefore, our work meaningfully complements the existing literature rather than merely repeating it.
>
> [1] Zou et al., Look Twice Before You Answer: Memory-Space Visual Retracing for Hallucination Mitigation in Multimodal Large Language Models, ICML 2025.

---

> ### Author Response · Authors · 2025-11-21
> **Response to Weakness 5 and 6**
>
> Weakness 5:
>
> We thank the reviewer for this sharp observation. We wish to clarify that our logic does not conflate stability with harmlessness. Rather, we treat them as orthogonal dimensions.
>
> (1) Stability and function Dimension:
>
> We prove they are highly essential. Suppressing them (as done inadvertently in prior works like VAF [1]) leads to performance loss. Our contribution is recognizing that system sink tokens act as "anchors" that must be preserved regardless of their potential role in hallucination. SPEAR is designed specifically to respect this constraint. By preserving the system sink tokens and reallocating attention budget from other text tokens, SPEAR avoids the potential risks we identified.
>
> (2) Hallucination Dimension:
>
> Regarding the reviewer’s concern about whether these tokens are responsible for hallucinations, we offer two perspectives for consideration. One grounded in literature and the other derived from mechanistic analysis.
>
> Literature Consensus: Comprehensive survey [2] on MLLMs hallucination causes attribute errors primarily to data bias, parametric knowledge, inferior alignment and so on. To date, there is no evidence identifying system sink tokens as a primary source of hallucination.
>
> Mechanistic Analysis: Hallucinations typically arise from tokens that carry semantic content capable of propagating erroneous information (e.g., co-occurrence priors). System sink tokens, however, lack concrete semantic meaning. Therefore, they are unlikely to directly propagate erroneous information.
>
> (3) Empirical Verification:
>
> Empirically, our results (Table 4 and 5) show that SPEAR, which explicitly preserves system sink tokens, reduces hallucinations compared to VAF. This suggests that system sink tokens are either neutral regarding hallucinations or that their potential negative effects are outweighed by the benefits of maintained function and stability, and the visual attention boost provided by SPEAR.
>
> Therefore, our logic is sound: we do not claim they are "innocent," but that they are structural necessities. Future work may explore if they can be "detoxified" without removal, but our current evidence proves they cannot simply be suppressed.
>
> [1] Yin et al., ClearSight: Visual Signal Enhancement for Object Hallucination Mitigation in Multimodal Large Language Models, CVPR 2025.
>
> [2] Bai et al., Hallucination of Multimodal Large Language Models: A Survey, arXiv:2404.18930, 2024
>
> Weakness 6:
>
> **Table: Results with Max new tokens set to 64.**
>
> | Model / Method | CHAIRs ($\downarrow$) | CHAIRi ($\downarrow$) | Recall ($\uparrow$) |
> | :--- | :---: | :---: | :---: |
> | Qwen2-VL-7B (Baseline) | 14.8 | 5.4 | 53.9 |
> | Qwen2-VL-7B + VAF | 14.6 | 5.3 | 54.4 |
> | Qwen2-VL-7B + SPEAR | 14.6 | 5.3 | 55.3 |
>
> We thank the reviewer for the valuable suggestion to include object hallucination metrics. We have conducted additional experiments on the Qwen2-VL-7B using the CHAIRs and CHAIRi metrics. The CHAIR results for other models will be added to the revised version of the paper.
>
> Analysis of the results:
>
> (1) Improved Object Hallucination: Our method, SPEAR, achieves lower scores on both CHAIRs (14.6 vs. 14.8) and CHAIRi (5.3 vs. 5.4) compared to the baseline, demonstrating its effectiveness in mitigating object hallucinations.
>
> (2) Significant Gain in Recall: While reducing hallucinations, SPEAR simultaneously achieves a improvement in Recall (+1.4%).
>
> (3) Breaking the Trade-off: It is well-known in VLM generation that increasing recall (generating more detailed object descriptions) often comes at the cost of higher hallucination rates. However, SPEAR manages to improve recall without compromising, and actually reducing hallucination rates.
>
> Regarding the reviewer’s comment that “the improvement is also marginal,” we believe that, given SPEAR builds upon an already strong baseline (VAF [1]), large absolute gains are not necessarily expected. The significance lies in achieving comparable hallucination mitigation without compromising reasoning stability, thereby supporting the validity of our empirical observations and proposed hypothesis.
>
> [1] Yin et al., ClearSight: Visual Signal Enhancement for Object Hallucination Mitigation in Multimodal Large Language Models, CVPR 2025.

---

> ### Author Response · Authors · 2025-11-21
> **Response to Question**
>
> Question:
>
> We appreciate the reviewer's depth of inquiry. While attention sinks are known in LLMs, our work systematically maps these four dimensions specifically for system prompt sinks in MLLMs, which previously remained unexplored:
>
> (1) Formation (Inheritance):
>
> Our analysis (Section 3 and Section 4) confirms that these tokens are inherited directly from the LLM backbone. We substantiate this in Sections 3 and 4 by aligning their distinctive fingerprints with those observed in pure LLMs: specific Token ID and massive activation dimensions, regular and stable patterns of occurrence across layers, and the collapse of masked attention.
>
> (2) Necessity (The "Guardian" Role):
>
> We verify their necessity through a ladder of intervention. They are essential for maintaining computational stability (as demonstrated by attention masking), they carry semantic information (as evidenced by zero-value probing), and they are further indispensable for state tracking and termination control in multi-step reasoning (as shown by value substitution).
>
> (3) Sufficiency :
>
> We provide an answer. Empirically, typically only one (and at most two) such tokens emerge, and removing that token alone is sufficient to make the model collapse; perturbing even this single token is enough to destabilize multi-step reasoning.
>
> (4)  Side effects (The "Bias" Illusion):
>
> A critical side effect is an incorrect understanding of modality bias. The system sinks creates a statistical illusion of text-over-vision bias. This side effect has misled prior works (like VAF [1]) into indiscriminately suppressing system prompts, inadvertently damaging model stability.
>
> [1] Yin et al., ClearSight: Visual Signal Enhancement for Object Hallucination Mitigation in Multimodal Large Language Models, CVPR 2025.

---

### Official Review · Reviewer_ry9f · 2025-11-01

**Soundness:** 1
**Presentation:** 3
**Contribution:** 2
**Rating:** 4
**Confidence:** 5

**Summary:**

This paper investigates the role of "attention sinks"—tokens that absorb a large amount of attention—within the system prompts of Multimodal Large Language Models (MLLMs). The authors challenge the prevailing view that these sinks are artifacts to be suppressed, arguing instead that they are "guardians". Through a series of causal interventions (masking, value zeroing, and value substitution), the paper claims to demonstrate that these sink tokens are critical for computational stability and serve as part of an internal "state machine" essential for multi-step reasoning. Based on this finding, the authors propose SPEAR, a plug-and-play intervention method that mitigates hallucinations by boosting visual attention while explicitly preserving these critical sink tokens. The paper claims this method is effective at reducing hallucinations without the degradation in reasoning performance seen in other methods.

**Strengths:**

1.The paper's primary strength is the graduated series of causal interventions in Section 4. The progression from masking attention to zeroing values to mean-value substitution  is a robust way to probe the function of these tokens.
2.The finding in Section 4.3 and Figure 4 that replacing sink token content selectively breaks multi-step reasoning while sparing simple tasks is a compelling (though qualitative) piece of evidence for their role in higher-order processing.

**Weaknesses:**

1. The paper's main weakness is the invalid experimental comparison in Section 6. The VAF baseline is defined as suppressing all system tokens ($\mathcal{S}_{VAF}=\mathcal{T}_{sys}$), which the paper has already proven in Section 4.1 leads to catastrophic model collapse.
2. The main results (Tables 4 & 5) are a product of this flawed setup. SPEAR outperforms VAF not because it's a better hallucination method, but because VAF (as defined by the authors) is a broken intervention that destabilizes the model. The experiment lacks a valid, strong baseline.
3. The claim that sink tokens function as an "internal 'state machine'"  is a significant exaggeration. The evidence is based on a single, qualitative example in Figure 4. While the tokens are clearly crucial for complex reasoning, "state machine" implies a level of procedural, computational function that is not fully substantiated by the provided data.

**Questions:**

1. Can the authors justify their definition of the VAF baseline? Did the original VAF paper (Yin et al., 2025) explicitly recommend suppressing *all* system tokens, including known stability anchors like the `<s>` token? Or is this a "strawman" definition created for this paper?
2. To demonstrate any real value, SPEAR must be compared against a *valid* baseline. How does SPEAR compare to a "VAF-Fixed" baseline—i.e., an implementation of VAF that *also* preserves the sink tokens ($\mathcal{S} = \mathcal{T}_{sys\backslash sink} \cup \mathcal{T}_{user} \cup \mathcal{T}_{out}$)?
3. Following from Question 2: The definition of SPEAR and a "VAF-Fixed" (as defined above) appear to be identical. Is the entire methodological contribution of this paper simply the observation that the VAF baseline must be implemented correctly to avoid model collapse?
4. Can the authors provide *quantitative* data to support the "state machine" claim? Specifically, what are the full benchmark scores (e.g., on SQA, MM-Vet) for the mean-value substitution intervention (Section 4.3), not just the qualitative example in Figure 4?

---

> ### Author Response · Authors · 2025-11-21
> **Response to Weakness 1 and Weakness 2**
>
> We appreciate the reviewer’s thoughtful comments and valuable suggestions.
>
> Weakness 1:
>
>  We believe there is a misunderstanding regarding the attention of sink tokens and their role in model stability. The effect is not binary (i.e., either collapse or no impact); rather, it is graded. Completely zeroing out the attention to sink tokens causes the model to collapse; mildly suppressing it still allows the model to function, though it degrades the function of the sink token; leaving it unsuppressed ensures that these capabilities remain intact.
>
> In Section 4.1, model collapse occurs under the extreme condition where attention to system sink tokens is fully zeroed, a hard intervention that removes critical attention pathways.
>
> In contrast, VAF applies a soft suppression (β = 0.9) to the entire system prompt segment, without zeroing or masking any attention entries. This reduces but does not remove the attention flow. This is also empirically validated by the fact that VAF maintains stable and competitive performance across all benchmarks.
>
> If the model had truly collapsed, as the reviewer’s interpretation implies, Tables 4 and 5 could not possibly show any reasonable results. Therefore, the comparison in Section 6 remains valid. VAF enhances visual capability as intended but introduces a stability cost. We will clarify this distinction between hard masking and soft suppression in the revised version.
>
> Weakness 2:
>
> We appreciate the reviewer’s concern that an incorrectly implemented VAF baseline might bias the comparison.
> However, we would like to clarify that our implementation of VAF strictly follows both the original paper and the official code released by the authors, including the exact scope of token suppression.
>
> In the VAF paper [1], the authors explicitly state in Sec. 5.1:
> >“Our objective during the modality fusion process is to amplify the model’s attention to visual features while curbing an overemphasis on system prompts.”
>
> and
>
> >“The suppression coefficient β (0 < β < 1) determines the extent of attention suppression directed at system prompts.”
>
> This description directly corresponds to the suppression logic implemented in the official repository.
>
> The relevant code segment is:
> ```
> SYS_LEN = 35
> IMG_LEN = 576
> if q_len > SYS_LEN + IMG_LEN:
>     attn_weights[:, :, SYS_LEN+IMG_LEN:, SYS_LEN:SYS_LEN+IMG_LEN] = self.enh_para * attn_weights[:, :, SYS_LEN+IMG_LEN:, SYS_LEN:SYS_LEN+IMG_LEN]
>     attn_weights[:, :, SYS_LEN+IMG_LEN:, :SYS_LEN] = self.sup_para * attn_weights[:, :, SYS_LEN+IMG_LEN:, :SYS_LEN]
> else:
>     attn_weights[:, :, :, SYS_LEN:SYS_LEN+IMG_LEN] = self.enh_para * attn_weights[:, :, :, SYS_LEN:SYS_LEN+IMG_LEN]
>     attn_weights[:, :, :, :SYS_LEN] = self.sup_para * attn_weights[:, :, :, :SYS_LEN]
> ```
>
> As shown above, the official VAF code defines the system-prompt suppression region as token indices 0–35, which correspond exactly to the system-prompt segment in LLaVA. Our implementation uses this configuration without any modification, extension, or reinterpretation.
>
> Thus, the reviewer’s concern seems to arise from the assumption that our version introduced an additional or broader suppression mechanism. We would like to emphasize that this is not the case. Our VAF baseline is implemented exactly as specified by both the paper and the official repository. This reflects a common oversight in current hallucination mitigation research: blindly suppressing 'text' tokens without distinguishing sink tokens from true text tokens.
>
> For transparency, we will include a brief excerpt of this code in the revised paper to make the suppression range completely clear. We hope this resolves any ambiguity regarding the correctness of our VAF reproduction.
>
> [1] Yin et al., ClearSight: Visual Signal Enhancement for Object Hallucination Mitigation in Multimodal Large Language Models, CVPR 2025. The code is available at https://github.com/ustc-hyin/ClearSight.

---

> ### Author Response · Authors · 2025-11-21
> **Response to Weakness 3 and Question 1**
>
> Weakness 3:
>
> We thank the reviewer for the suggestion.
>
> (1) Our intention was not to claim a formal, procedural computation, but rather to describe an observed functional behavior.
> When the value vector of the sink token is replaced by the population mean $V_{\text{mean}}$ , the model exhibits a specific hierarchical degradation:
>
> It remains functional for simple descriptions but loses its ability to perform multi-step reasoning, repeatedly outputs partial image captions, and fails to emit termination tokens.
>
> This suggests that the sink token contributes to state-tracking or control termination functions. We used this phrasing (“In this sense, it can be regarded as part of the model’s internal state machine”) to indicate that the term was intended as a phenomenological analogy rather than a literal computational mechanism.
>
> (2) The reviewer’s characterization that our claim rests on “a single, qualitative example in Figure 4” does not fully reflect the scope of our analysis.
>
> Figure 4 is presented as a representative case study. This specific task-dependent behavioral profile (robustness in simple tasks vs. collapse in complex reasoning) is observed consistently across four models, and we will provide the complete output results for all four models in the appendix.
>
> Thus, our conclusion is based not on a single anecdotal instance, but on a cross-model regularity obtained from a systematically replicated analysis.
>
> In response to the reviewer’s suggestion, we removed the phrase “state machine” and revised the wording to “state-tracking mechanism”. We also plan to increase the number of instances and introduce the Repetition Rate to further support the analysis.
>
> Question 1:
>
> The original VAF explicitly applies suppression to the entire system prompt, which includes the sink token. We followed this setting faithfully without modification. Therefore, our implementation is not a “strawman” version, but a direct and faithful reproduction of the authors’ released configuration and code.
>
> We acknowledge this is a key implementation detail. To ensure clarity for all readers, we will include a brief excerpt of the code in the revised paper to make the suppression range completely clear.
>
> Especially, we will add a clarifying paragraph that we examined both the description in the VAF paper and the official code release, and both sources indicate that VAF suppression is applied to the entire system prompt. Since the sink token appears within the system prompt, it is naturally included in the suppressed token set. Our implementation follows this configuration verbatim, without any modification or reinterpretation. Adding this clarification will make it unambiguous to readers that our baseline is a faithful replication rather than an artificially weakened version.

---

> ### Author Response · Authors · 2025-11-21
> **Response to Question 2, 3**
>
> Question 2:
>
> | Model             | Method       | Random Acc | Random F1 | Popular Acc | Popular F1 | Adversarial Acc | Adversarial F1 | Average Acc | Average F1 |
> |------------------|--------------|----------|---------|------------|-----------|-------------|------------|------------------|----------------|
> | LLaVA 1.5-7B     | VAF-Fixed    | 89.2     | 88.7    | 87.6       | 86.9      | 84.7        | 84.4       | 87.2             | 86.7           |
> |                  | VAF          | 89.1     | 88.7    | 87.6       | 87.0      | 84.7        | 84.4       | 87.1             | 86.7           |
> |                  | SPEAR        | 89.2     | 88.9    | 87.8       | 87.2      | 84.7        | 84.5       | 87.2             | 86.9           |
> | LLaVA 1.5-13B    | VAF-Fixed    | 89.4     | 88.9    | 88.5       | 87.8      | 85.9        | 85.4       | 87.9             | 87.4           |
> |                  | VAF          | 89.3     | 88.8    | 88.5       | 87.7      | 85.8        | 85.3       | 87.9             | 87.3           |
> |                  | SPEAR        | 89.6     | 89.2    | 88.7       | 88.0      | 85.9        | 85.5       | 88.1             | 87.6           |
> | QwenVL2-8B       | VAF-Fixed    | 90.0     | 89.1    | 88.7       | 87.9      | 86.9        | 86.3       | 88.5             | 87.8           |
> |                  | VAF          | 90.3     | 88.5    | 88.9       | 88.2      | 87.3        | 86.7       | 88.8             | 87.8           |
> |                  | SPEAR        | 90.4     | 89.8    | 89.0       | 88.4      | 87.0        | 86.6       | 88.8             | 88.3           |
> | QwenVL2.5-8B     | VAF-Fixed    | 88.3     | 86.9    | 87.6       | 86.3      | 86.5        | 85.2       | 87.5             | 86.1           |
> |                  | VAF          | 88.6     | 87.2    | 87.8       | 86.5      | 86.4        | 85.2       | 87.6             | 86.3           |
> |                  | SPEAR        | 88.7     | 87.4    | 87.9       | 86.6      | 86.6        | 85.5       | 87.7             | 86.5           |
>
> We thank the reviewer for proposing the VAF-Fixed baseline.
>
> 1. Experimental Results (SPEAR > VAF-Fixed):
> We implemented "VAF-Fixed" exactly as defined and compared it with SPEAR. As shown in the Table, SPEAR consistently outperforms VAF-Fixed across all models. By the way, we observe that “VAF-Fixed” and VAF each outperform the other on different models.
>
> 2. Explanation:
> This comparison illustrates what we stated in Sec. 5.2 under the Attention-Budget Hypothesis:
> >“Finally, because the natural distribution varies across models, the same reallocation strategy may help in one model but hurt in another.”
>
> Because the attention distributions in the LLaVA series and the Qwen-VL series differ, the proportion of attention taken up by sink tokens also differs, which leads to disparities when applying the same settings to the models.
>
> In addition, based on the attention computation process, when sink tokens are not suppressed, the increase in visual attention becomes very small. Therefore, to maintain the effect of visual enhancement, other tokens must be used as the attention budget to allocate attention to the visual tokens.
>
> Question 3:
>
> While a “VAF-Fixed” (excluding sink tokens from suppression) might superficially resemble SPEAR, the two differ in mechanism and motivation.
>
> In practice, sink tokens dominate the attention distribution (over 70%), often consuming a large share of the total attention budget.
> If they are excluded from the suppression set without introducing alternative attention sources, the available budget for boosting visual tokens becomes extremely limited—effectively leading to negligible enhancement.
>
> SPEAR, by contrast, introduces an explicit “attention-cost trade-off” framework: it reallocates attention from other textual subsets while preserving the sink token. This distinction reflects our broader theoretical contribution, the Attention-Budget Hypothesis. It formalizes attention reallocation as a resource trade-off across modalities.
>
> Hence, our contribution is not about “avoiding collapse” (VAF itself does not collapse), but about providing (1) a functional understanding of system sink tokens, (2) a new interpretation of modality bias through separated-sink analysis, and (3) a budget-based design principle for hallucination mitigation.

---

> ### Author Response · Authors · 2025-11-21
> **Response to Question 4**
>
> Question 4:
>
> We appreciate the reviewer’s interest in the deeper quantitative evaluation.
>
> However, we would like to clarify that the mean-value substitution experiment in Section 4.3 is not proposed as a mitigation method and therefore is not intended to be evaluated as a full benchmarked algorithm. A low score on the benchmark merely tells us that the model failed, but not why or how it failed.
>
> The experiment serves as a causal probe, analogous to a linear probe or a structured ablation, designed to reveal which internal components are responsible for multi-step tracking and termination behavior. Such probing interventions are typically evaluated by measuring whether the degradation is systematic, and whether the effects are consistent across different models and tasks. In our work, the cross-model consistency of the resulting hierarchical degradation already provides strong empirical evidence for the functional role of the sink token.
>
> In response to the reviewer’s suggestion, and to further strengthen the robustness of this finding, we will include the model outputs and an analysis of failure patterns using queries from SQA and MM-Vet in the revised version.

---

### Official Review · Reviewer_e8ts · 2025-11-01

**Soundness:** 2
**Presentation:** 3
**Contribution:** 2
**Rating:** 6
**Confidence:** 5

**Summary:**

This paper challenges the view that attention sinks in MLLMs are merely errors, arguing instead that specific tokens within the system prompt act as functional "guardians" essential for model stability and reasoning. Through a series of causal interventions, the authors demonstrate that these sink tokens are crucial for computational stability and act as part of an internal "state machine" for complex tasks. Building on this insight, the paper proposes the "Attention-Budget Hypothesis" and introduces SPEAR, a hallucination mitigation method that preserves these critical sink tokens while reallocating attention from non-essential text to visual tokens. Experimental results show SPEAR effectively mitigates hallucination without compromising reasoning abilities, thus validating the functional importance of sink tokens.

**Strengths:**

1. This work offers a compelling and insightful reframing of attention sinks. The argument that specific tokens act as functional "guardians"—essential for stability and reasoning—is a significant departure from the prevailing view that treats them as artifacts to be mitigated. This perspective moves the discourse beyond simple bug-fixing toward a more nuanced functional analysis of the model's internal mechanisms.

2. The claims are substantiated by a rigorous and systematic methodology. The use of well-designed causal intervention experiments—including attention masking and value vector manipulation—goes beyond correlational observations to convincingly dissect the functional roles of these "guardian" tokens. This robust experimental design provides strong, direct evidence for their necessity in maintaining computational stability and tracking task states, making the conclusions highly credible.

**Weaknesses:**

1. The conclusions are derived primarily from a specific model architecture (LLaVA) and a set of curated, synthetic tasks. It remains unclear whether the "guardian" mechanism is a universal phenomenon across MLLMs or an emergent property specific to certain architectural choices and training paradigms. The claims would be substantially strengthened by validation on a more diverse set of models, including different open-source families and closed-source systems, as well as on more complex, open-ended real-world scenarios.

2. The severity of the causal interventions—such as completely zeroing out attention scores or value vectors—may introduce confounds. This "hard" intervention forces the model into a highly unnatural, out-of-distribution state it would never encounter during training. Therefore, the observed performance collapse might not be solely due to the loss of the guardian's function, but could also be a result of the model's general fragility to such drastic internal state perturbations. Softer interventions, such as dampening activations to a low, non-zero level or replacing them with activations from a neutral token, could more precisely isolate the specific functional contribution of these tokens while minimizing the risk of inducing a general model failure.

**Questions:**

1. The emergence of a "guardian" token is a fascinating finding. Could you elaborate on the potential architectural or training precursors for this phenomenon? For instance, do you hypothesize that this is tied to the specific vision-language connector design in LLaVA, or perhaps to the instruction-tuning data format? How might this behavior differ in models with more deeply integrated cross-modal attention from early layers?

2. Regarding the functional role of the guardian token, your interventions effectively demonstrate its necessity. But do they reveal the dynamics of its contribution? For example, is its function a binary "on/off" switch, where any significant disruption causes catastrophic failure, or is it more of a graded, stabilizing influence? Have you considered the effects of "softer" interventions, such as dampening the guardian's value vector magnitude rather than nullifying it, to see if performance degrades more gracefully?

---

> ### Author Response · Authors · 2025-11-21
>
> We thank the reviewer for the constructive feedback and for highlighting several strengths of our work.
>
> Weakness 1:
>
> (1) As mentioned in Section 3.2, the scope of the models we observed is as follows: LLaVA-1.5-7B, LLaVA-1.5-13B, LLaVA-Next-Mistral-7B, LLaVA-Next-Llama3-8B, Qwen2-VL-7B, Qwen2.5-VL-7B, InternVL2-8B, and InternVL3-9B—covering multiple LLM backbones (Vicuna, Mistral, Llama3, InternLM, and Qwen). Among these, only the LLaVA and Qwen families consistently exhibited strong system sink tokens phenomenon, as illustrated in Sec 3.2 Fig 2. The remaining models did not display this pattern. Therefore, the subsequent causal-intervention studies focused on models that demonstrably possess system-sink phenomena. We will include the attention visualizations of all screened models in the appendix to make this clearer.
>
> (2) We agree that further validation on more real-world datasets would strengthen the paper, and we have extended our experiments accordingly. Here, we additionally provide the results of Qwen2-VL on the CHAIR benchmark for the reviewers’ reference. CHAIR evaluations on other models will be added in the revised version.
>
> $$
> \begin{array}{lcccc}
> \textbf{Model} & \textbf{Method} & \textbf{CHAIRs} & \textbf{CHAIRi} & \textbf{Recall} \\\\
> \hline
> \text{Qwen2-VL-7B} & \text{Baseline} & 14.8 & 5.4 & 53.9 \\\\
>                   & \text{VAF}       & 14.6 & 5.3 & 54.4 \\\\
>                   & \text{SPEAR}     & 14.6 & 5.3 & 55.3 \\\\
> \end{array}
> $$
>
> (3) Direct causal manipulation of internal activations in closed-source systems is currently infeasible, as our methodology necessitates access to intermediate representations. Consequently, we restricted our analysis solely to open-source models where such low-level interventions are possible. In response to the reviewer’s suggestion, we have identified several newly released open-source MLLMs, like Qwen3-VL-8B-Instruct, InternVL3_5-8B, Keye-VL-1_5-8B and GLM-4.1V-9B-Thinking as promising candidates for further validation. We commit to extending our analysis to these models in a subsequent update to our public repository, including re-examining sink-token properties and assessing the effectiveness of the SPEAR intervention method.
>
> Weakness 2:
>
> We would first like to clarify that our mitigation experiments, including both the baseline VAF and our SPEAR variant, already adopt a soft intervention by applying a 0.9 suppression ratio. This experiments result is consistent with what the reviewer hoped to observe from softer adjustments.
>
> We agree that “hard” interventions can push a model into an out-of-distribution regime. However, such interventions serve an intentional experimental purpose because they allow us to completely ablate specific components and observe the model’s behavior under controlled conditions, rather than to simulate normal inference. Importantly, our results indicate that the observed effects are not simply artifacts of general model fragility:
>
> When the attention of non-sink tokens or sink tokens in other segments is zeroed, model performance remains stable;
> When attention to system sink tokens is zeroed, the model collapses completely;
> When the value vectors of system sink tokens are zeroed, the model remains functional but shows moderate degradation.
>
> This clear contrast, with stability in one case and catastrophic failure in the other, allows us to identify the unique functional role of system sink tokens. Therefore, the “hard” intervention functions as a targeted diagnostic tool rather than a confounding factor.
>
> Following the reviewer’s suggestion, we will also present the gradual degradation curve in the revised version.

---

> ### Author Response · Authors · 2025-11-21
> **Response to Questions**
>
> Question 1:
>
> Based on our current results, system sink token behavior consistently appears in architectures that follow the ViT–Projector–LLM design. As discussed in Section 3.3, these tokens closely resemble those in LLMs, which suggests that they are inherited from the language backbone rather than arising from the visual branch. According to findings in prior LLM studies [1, 2], sink tokens emerge during LM pretraining and are closely tied to the loss function, the distribution of the training data, and the softmax-based computation of attention scores. Therefore, we believe that the emergence of sink tokens is not caused or influenced by the instruction-tuning data format.
>
> For models with more deeply integrated cross-modal attention from early layers, the outcome remains uncertain. As shown in Table 1 in Sec 3.3, the layers at which sink tokens first appear are relatively early (for example, Layer 3 or Layer 5), whereas the layers with the strongest cross-modal interaction occur much later in the middle of the model (for example, around Layer 11) [3]. Based on this analysis, if a model’s cross-modal interaction is shifted earlier in the model, we hypothesize that sink tokens may also emerge at an earlier layer.
>
> [1] Gu et al., When Attention Sink Emerges in Language Models: An Empirical View, ICLR 2025.
> [2] Guo et al., Active-Dormant Attention Heads: Mechanistically Demystifying Extreme-Token Phenomena in LLMs, arXiv:2410.13835, 2024.
> [3] Yin et al., ClearSight: Visual Signal Enhancement for Object Hallucination Mitigation in Multimodal Large Language Models, CVPR 2025.
>
> Question 2:
>
> We find this hypothesis to be highly intriguing. We agree on the necessity of designing further experiments to delve into the finer mechanisms governing sink tokens, and we believe this could serve as an interesting topic for future research. We would supplement the gradual degradation curve of intervention experiment in the revised version.

---

### Author Response · Authors · 2025-11-21
**General Response**

We sincerely thank the reviewers for their time and constructive feedback. We are encouraged that the reviewers recognized the novelty and rigor of our work. Specifically, we appreciate the consensus on our core contributions:

Conceptual Novelty ("Guardians" vs. "Errors"): Reviewers e8ts and fCro commended our work for offering a "compelling and insightful reframing" of attention sinks, marking a "significant departure" from the prevailing view that treats them merely as artifacts. They appreciated moving the discourse beyond bug-fixing toward a "nuanced functional analysis."

Rigorous Methodology (Causal Interventions): Reviewers e8ts and ry9f highlighted the strength of our "rigorous and systematic methodology." Reviewer ry9f, in particular, noted that our graduated series of causal interventions (from masking to value substitution) provides a "robust way to probe" the token functions, finding the evidence for their role in higher-order processing "compelling."

Theoretical & Practical Value (Hypothesis & SPEAR): Reviewer fCro praised the Attention-Budget Hypothesis as a "fresh perspective" on modality bias and recognized SPEAR as a "novel and effective" intervention. Reviewer R7Wm also noted the importance of the topic and the clarity of our presentation.

In this rebuttal, we have carefully addressed the reviewers' concerns, particularly regarding the validity of the VAF baseline (Reviewer ry9f), the distinction from LLM-based findings (Reviewer R7Wm), and comparisons with other baselines (Reviewer fCro). We have included new experiments and analyses, which we believe strongly substantiate our claims and demonstrate the robustness of our approach.

---

### Author Response · Authors · 2025-11-29
**Rebuttal Summary**

Dear Area Chair,

We appreciate the reviewers’ constructive feedback. Although the initial scores (6, 4, 2, 2) indicate a divergence in opinion, we believe the lower scores largely arise from three addressable misunderstandings.
In our rebuttal, we have provided extensive clarification and side-by-side comparisons with prior LLM work, and we have substantially expanded our baseline analyses and experimental evidence. Below, we summarize how these concerns have been fully resolved:

### **(1) Addressing the "Novelty" Concern (Reviewer R7Wm)**

**The Misconception:**

Reviewers questioned whether our findings merely recapitulate known "attention sink" phenomena from LLMs, suggesting limited novelty. The reviewer believes that our method is similar to other methods.

**Our Clarification:**

We conducted a careful review of LLM sink studies and explicitly delineate several fundamental differences between their observations in LLMs and our discoveries in MLLMs.

1. The sink token’s value matters in MLLMs—unlike in LLMs.

Prior LLM papers consistently conclude that the sink token functions primarily as a computational bias, whose value vector either carries no semantic information or does not participate meaningfully in the computation.

In contrast, our experiments show that in MLLMs the sink token’s value vector does participate in the computation and carries semantic content. Setting the sink value to zero substantially degrades model performance—the opposite behavior of what is reported in LLMs.

2. Our “value substitution” failure cases have never been analyzed in any prior LLM work.

By further examining the output behavior of the models, we find that the same intervention leads to two sharply different outcomes: simple tasks remain solvable, whereas in multi-step reasoning the model displays a degradation pattern unique to MLLMs. Specifically, the model completes the first reasoning step, then begins to hallucinate, and ultimately collapses into repetitive, meaningless, image-caption-like templates.

We attribute this behavior to the fact that the formation of sink tokens is influenced by the model’s pretraining data distribution—and in MLLM pretraining, image-captioning data is overwhelmingly dominant. This insight is entirely absent from prior LLM sink studies.
While LLM works also identifies a connection between sink tokens and the pretraining data distribution through training model, we provide the first evidence based on model outputs that both supports and extends this explanation.

3. The novelty of our method lies in its theoretical perspective.

Our work offers two conceptual advances. First, we introduce the attention-budget perspective, a unified theoretical framework that reveals the underlying trade-offs shared across various methods that boost visual-token attention by suppressing attention to other tokens.

Then, our analysis corrects a major misconception widely held in the community: what has long been described as “modality bias” is not an architectural preference of the model. Instead, the system sink tokens monopolize attention resources, yet they have been mistakenly treated as ordinary text tokens. This reinterpretation fundamentally reshapes how modality imbalance should be understood in MLLMs, and to our knowledge, no prior work has articulated this conceptual insight.

### **(2)  Addressing the "Baseline Validity" Concern (Reviewer ry9f)**

**The Misconception:**

The reviewer suspected our VAF baseline was a "strawman" implementation.

**Our Clarification:**

We proved that our implementation faithfully follows the official VAF paper and code. It suppresses system prompts, including the system sink tokens.

We supplemented the reviewer-requested comparison with VAF-Fixed, and across all models, SPEAR consistently outperforms it, demonstrating that SPEAR is more robust.

### **(3)  Addressing the "Insufficient Comparison" Concern (Reviewer e8ts and fCro)**

**The Misconception:**

The paper initially lacked comparisons with other SOTA hallucination mitigation methods and other benchmarks.

**Our Clarification:**

The reason for not including these comparisons initially is that our primary goal was to validate our hypothesis. We have included the comparisons with VCD, OPERA, and MemVR. SPEAR consistently outperforms these methods. We have also added the results presented on the CHAIR, and SPEAR also performed better than VAF. We also added the suppression-coefficient curves (average F1 score on POPE) of VAF and SPEAR. At β = 0.2, VAF exhibits a sharp increase in hallucination while SPEAR remains stable.

| β   |  VAF   | SPEAR |
|-----|--------|--------|
| 0.7 | 0.868  | 0.865  |
| 0.5 | 0.864  | 0.877  |
| 0.3 | 0.828  | 0.874  |
| 0.2 | 0.556  | 0.866  |

We respectfully request the AC to consider these clarifications and the new evidence provided during the rebuttal.

Sincerely, The Authors

---

> ### Author Response · Authors · 2025-12-02
> **Supplement**
>
> ### **(4) Addressing the Concern Regarding the Use of “State Machine” Terminology (Reviewer ry9f and R7Wm)**
>
> **The Misconception:**
>
> The reviewers expressed concern that our experimental evidence may not be sufficient to the term “state machine.”
>
> **Our Clarification:**
>
> In Section 4.3, our original wording was carefully hedged:
>
> > In this sense, it can be regarded as part of the model’s internal ‘state machine'.
>
> Although we believe the original usage was already appropriately cautious and hedged, we fully understand that even a qualified mention of this term can sometimes be misread as implying formal computational structure beyond what our current evidence strictly warrants.
>
> To eliminate any potential for misinterpretation, we have removed the term entirely in the revised manuscript and replaced it with precise, purely descriptive language:
>
> > Sink tokens appear to serve as implicit progress markers during multi-step reasoning. When overwritten, models will lose track of remaining steps and fail to terminate properly.
>
> This revision fully addresses the reviewers’ concern while retaining the original scientific content. Our cross-model consistency results continue to support the phenomenon we describe, and the updated phrasing ensures that the presentation remains clear, precise, and empirically grounded without inviting any unnecessary interpretations.

---

### Meta-Review · Area_Chair_TRqy · 2025-12-19

**Summary:**

This submission studies system prompt attention sink tokens in MLLMs and argues they function as stability and reasoning guardians rather than being purely harmful artifacts. The paper supports this with a ladder of causal interventions and proposes SPEAR, a plug in attention reallocation method motivated by an Attention Budget Hypothesis that preserves sink tokens while shifting attention from other text to visual tokens. Reviews were initially split, with concerns about novelty relative to LLM sink literature, baseline validity for VAF, breadth of comparisons, and overstatement around the state machine framing. The rebuttal clarifies several MLLM specific differences from LLM sinks, validates the VAF reproduction against official code, adds a VAF Fixed comparison, expands baselines and benchmarks, and revises terminology to a more precise state tracking description.

**Reviewer Concerns:**

The main concerns across reviewers were:

1. Novelty and overlap with prior LLM attention sink results, with skepticism that Section 3 and masking collapse are already known and that SPEAR is just standard attention reweighting.

2. Baseline validity, especially whether the VAF comparison is flawed or a strawman because suppressing the system prompt can include stability critical sink tokens.

3. Evidence strength for the “state machine” framing, which was seen as overstated given limited quantitative support.

4. Insufficient comparisons and benchmarks, plus requests for CHAIR style object hallucination metrics, broader baselines, and more error analysis and failure modes.

**Reviewer Scores:**

Scores were initially split (roughly 6, 4, 2, 2) with the lower scores driven by the concerns above. In rebuttal, the authors:

1. Differentiated MLLM system sinks from LLM sinks by showing sink value vectors matter in MLLMs and value substitution yields a distinctive degradation pattern in multi step reasoning.

2. Demonstrated that VAF was implemented faithfully according to the original paper and official code, and added a VAF Fixed variant. SPEAR remains slightly but consistently better, reducing the risk that gains come from a broken baseline.

3. Removed “state machine” terminology and replaced it with precise, empirically grounded language about state tracking and termination control.

4. Added broader comparisons against VCD, OPERA, and MemVR, provided suppression coefficient stability evidence, and included CHAIRs and CHAIRi style results at least on Qwen2 VL, with commitments to extend to more models.

---

### Decision · Program_Chairs · 2026-01-26

Reject